# Genome-Wide Analysis of Sweet Potato Ammonium Transporter (AMT): Influence on Nitrogen Utilization, Storage Root Development and Yield

**DOI:** 10.3390/ijms242417424

**Published:** 2023-12-13

**Authors:** Ya-Yi Meng, Ning Wang, Hai-Yan Zhang, Ran Xu, Cheng-Cheng Si

**Affiliations:** 1Collaborative Innovation Center of Nanfan and High-Efficiency Tropical Agriculture, Hainan University, Sanya 572025, China; 21220951310162@hainanu.edu.cn (Y.-Y.M.); xuran@hainanu.edu.cn (R.X.); 2Key Laboratory of Quality Regulation of Tropical Horticultural Crop in Hainan Province, School of Tropical Agriculture and Forestry (School of Agricultural and Rural, School of Rural Revitalization), Hainan University, Danzhou 571700, China; nwanghnu@163.com; 3School of Breeding and Multiplication (Sanya Institute of Breeding and Multiplication), Hainan University, Sanya 572025, China; 4Scientific Observation and Experimental Station of Tuber and Root Crops in Huang-Huai-Hai Region of Agriculture Ministry, Crop Research Institute, Shandong Academy of Agricultural Sciences, Ji’nan 250100, China; zhanghaiyan@saas.ac.cn

**Keywords:** field experiments, *IbAMT1.3*, *IbAMT1.5*, N deficiency, N uptake efficiency

## Abstract

Ammonium, as a major inorganic source of nitrogen (N) for sweet potato N utilization and growth, is specifically transported by ammonium transporters (*AMTs*). However, the activities of *AMT* family members in sweet potatoes have not been analyzed. In the present study, the sweet potato cultivar ‘Pushu 32’, which is planted in a large area in China, was used in field experiments at the Agricultural Base of Hainan University (20°06′ N, 110°33′ E) in 2021, and Sanya Nanfan Research Institute of Hainan University (18°30′ N, 109°60′ E) in 2022. Four N levels were tested: 0, 60, 120, and 180 kg ha^−1^. The results are as follows. Twelve *IbAMT* genes were identified in the sweet potato genome, which were classified into three distinct subgroups based on phylogeny; the same subgroup genes had similar properties and structures. *IbAMT1.3* and *IbAMT1.5* were mostly expressed in the storage roots under N deficiency. Compared with the NN and HN groups, *IbAMT1.3* and *IbAMT1.5* expressions, N content in storage roots, N uptake efficiency at the canopy closure, N fertilization contribution rates, number of storage roots per plant, storage root weight, and yield were all increased in the MN group. Furthermore, there was a significant positive correlation between the expressions of *IbAMT1.3* and *IbAMT1.5* with N content in the storage roots of sweet potato. In a word, *IbAMT1.3* and *IbAMT1.5* may regulate N utilization, affect the development of the storage root. and determine the yield of sweet potato. The results provide valuable insights into the *AMT* gene family’s role in the use of N and effects on storage root development and yield in sweet potatoes.

## 1. Introduction

Sweet potato (*Ipomoea batatas* (L.) Lam.) yield is determined by storage root development, and nitrogen (N) is the key nutrient regulating storage root development [1]. A too high or too low application of N is not conducive to storage root formation: N deficiency, which reduces dry matter accumulation, is not conducive to storage root formation and reduces the yield of sweet potato [2], although N use efficiency is higher under N deficiency condition [3]. Applying the appropriate amount of N is beneficial for increasing the number of storage roots, the dry matter accumulation in storage roots, and the yield of sweet potato [3,4,5]. Excessive N application leads to the excessive growth of shoots [6], increased plant N accumulation [7,8,9], and increased dry matter accumulation in the whole plant, but to delayed storage root formation, significantly reduced N use efficiency (NUE), and decreased yield of sweet potato [10,11,12]. Therefore, proper N application is an effective measure to promote storage root formation and increase the yield of sweet potato.

Both ammonium and nitrate are crucial for plant growth and yield. Ammonium requires less energy to assimilate; therefore, ammonium nitrogen is preferentially absorbed [13] and is considered as a superior nitrogen source [14]. Urea is the most commonly used N fertilizer because of its low cost, high N content, and rapid N release [15]; it can be hydrolyzed into ammonium (NH_4_^+^) by urease [16]. The transportation of NH_4_^+^ by the roots from the soil is the start of ammonia assimilation and the foundational step for N utilization [17]. The ammonium transporter (AMT) exists widely in plants, which is a kind of plasma membrane protein that exclusively transports NH_4_^+^ [18]. The first ammonium transporter *AtAMT1.1* was isolated from the model plant *Arabidopsis* through growth complementarity [19]. Subsequently, several *AMT* genes have been identified in Arabidopsis, rice, tomato, and other plant species [20,21,22]. Plant *AMT* genes form a polygenic family, divided into *AMT1s* and *AMT2s* subfamilies according to gene homology [23]. For example, there are five *AMT1* genes and one *AMT2* gene in *Arabidopsis thaliana* [24]. Among them, *AtAMT1.2*, *AtAMT1.3*, *AtAMT1.5*, and *AtAMT2* were found to be mainly expressed in the roots; *AtAMT1.4* showed pollen-specific expression; and *AtAMT1.1* was expressed in roots, stems, and leaves [25,26]. In rice, *OsAMT1;2* and *OsAMT1;3* are mainly expressed in the roots, while *OsAMT1;1*, *OsAMT2;1*, and *OsAMT3;1* are expressed in both the roots and shoots [19]. In tomatoes, *SlAMT1-1* and *SlAMT1-2* were found to be mainly expressed in roots and leaves and *SlAMT1-3* in seedlings and leaves [27]. *AMT* genes are differentially expressed. Additionally, the expression of the *AMT* gene is affected by nitrogen levels and nitrogen forms. Studies on cassava showed that the expression of all *MeAMT1* genes in roots was upregulated under low NH_4_^+^ (NH_4_Cl) stress [28]. After rape seedlings that were cultured in water for 10 days under N (NH_4_Cl)-suitable conditions were transferred to a N (NH_4_Cl)-deficient environment for 5 days, it was found that the expression of all *AMT* genes in the roots was upregulated [29]. In rice, the transcript levels of *OsAMT1.1* in roots exhibited a several-fold decrease within 48 h when plants acclimated to 10 μM external NH_4_^+^ ((NH_4_)_2_SO_4_) for 3 weeks were transferred to 10 mM NH_4_^+^ ((NH_4_)_2_SO_4_) [30]. This suggests that increased NH_4_^+^ concentrations significantly reduce the expression of *OsAMT1.1*. Compared with ammonium, nitrate can inhibit the absorption of ammonium salts, resulting in a decrease in the expression of *OsAMT* genes (*OsAMT1;1*, *OsAMT1;2*, and *OsAMT1;3*) under ammonium–nitrate mixed conditions [31]. Studies on *Arabidopsis* revealed that ammonium transporters, such as *AtAMT1.1 AtAMT1.3* and *AtAMT2*, are transcriptionally regulated by urea [32]. Therefore, it can be concluded that urea has the ability to affect the activity of *AMT* genes, and the upregulation of these genes can enhance the plant roots’ ability to uptake NH_4_^+^. However, compared to other crops, there is a lack of research on the *AMT* gene family in sweet potatoes and the function of *IbAMT* transporters in NH_4_^+^ absorption and utilization. Additionally, it remains unclear whether the expression of *IbAMTs* is related to sweet potato storage root development and yield under different nitrogen treatments.

To address this gap in knowledge, this study aimed to identify the members of the *AMT* gene family in sweet potatoes and screen for key *IbAMT* genes that may regulate N utilization and impact storage root development and yield in sweet potatoes. The findings of this study provide a theoretical basis for further investigation into the function and mechanism of *IbAMT* regulating N uptake and utilization in sweet potatoes.

## 2. Results

### 2.1. Yield and Agronomic Traits at Harvest

The effect of N on yield was similar in both years (Table 1). The number of storage roots per plant and storage root weight and yield increased first and then decreased with increasing N application, with the highest value recorded in the MN treatment. The difference between MN and NN was significant (*p* < 0.05).

In Table 2, the results of two years of field experiments showed that the leaf, petiole, and stem weight increased significantly (*p* < 0.05) with increasing N application. Storage root weight per plant increased first and then decreased with increasing N application, with the peak recorded in the MN treatment. The difference between MN and NN was significant (*p* < 0.05).

### 2.2. Agronomic Traits at Canopy Closure

Figure 1 shows the sweet potato in the MN treatment grew best among the four N levels (Figure 1). The weight of the leaves, petioles, and stems increased first and then decreased with the increase in N application and reached a peak in MN in 2021 and 2022 (Figure 2). Compared with NN, these values were significantly higher in MN (*p* < 0.05).

In both years at canopy closure, with the increase in N application, number of storage roots per plant, storage root weight, and storage root weight per plant increased first and then decreased (Table 3), and reached the peak in the MN treatment. There was a significant difference between MN and NN (*p* < 0.05).

### 2.3. N Content

In 2021 and 2022 (Figure 3), at canopy closure, with increasing N application, the N content in the leaf, petiole, stem, and storage root increased first and then decreased, reaching the maximum in MN, which was significantly different from NN (*p* < 0.05). At harvest, the N content in the leaf, petiole, and stem increased with the increase in the N application rate, and the difference was significant between MN and NN (*p* < 0.05). The N content in the storage root increased first and then decreased with the increase in N application, reaching the maximum in MN, which was significantly different from NN (*p* < 0.05).

### 2.4. N Use Efficiencies

N uptake efficiency increased first and then decreased with the increase in N application, reaching a peak in the MN treatment, and MN was significantly different from NN at canopy closure (*p* < 0.05). However, N uptake efficiency significantly decreased with the increase in N application at harvest time (*p* < 0.05). N use efficiency significantly decreased with the increase in N application (*p* < 0.05). N fertilization contribution rates increased first and then decreased with the increase in N application and reached a peak in the MN treatment. MN was significantly different from NN (*p* < 0.05) (Table 4).

### 2.5. Genome-Wide Identification

After removing redundant entries, 12 *AMT* genes with the specific domain were identified from the sweet potato genome database via a BLAST search and TBtools software, from *IbAMT1.1* to *IbAMT1.5* and *IbAMT2.1* to *IbAMT2.7* based on their location on the chromosomes (Appendix A) and the result of the phylogenetic tree (Figure 4). The protein sequences of the IbAMT protein comprised 327–718 amino acids (aa), and the predicted molecular weights varied from 35,229.26 (*IbAMT2.6*) to 76,762.08 Da (*IbAMT1.3*). The range of the theoretical isoelectric point (pI) was 5.94 (*IbAMT1.1*)—8.77 (*IbAMT2.2*), and the instability index ranged from 23.74 (*IbAMT1.4*) to 45.62 (*IbAMT1.3*). Furthermore, subcellular localization prediction showed that most of the *IbAMT* gene family members are located on the plasma membrane (Table 5).

### 2.6. Phylogenetic Analysis

The determined full-length amino acid sequences from seven different species were aligned using ClustalW, and then a phylogenetic analysis was performed based on the multiple alignments results. The phylogenetic tree shows that the AMT proteins are divided into two subfamilies, and AMT2 can be further divided into two clusters, while AMT1 members are all concentrated in AMT1 (Figure 4), confirming that the *AMT1* gene has were more conserved than the *AMT2* gene during evolution.

### 2.7. Gene Structure and Motif Composition

The structure of IbAMT proteins and genes were further analyzed, including the *IbAMT* conserved motif and the conserved domain according to the evolutionary relationship (Figure 5). In the phylogenetic tree, the similarities between motif compositions, gene structures, and similarity alignment of AMT proteins in the same subgroup suggest that *IbAMT* genes were conserved during evolution (Figure 5, Appendix A). Five conserved motifs were identified, and motif 2 is highly conserved in the *IbAMT* family, whereas motifs 5 are present only in the *IbAMT1* subfamily (Figure 5B and Appendix A). In Figure 5C, *IbAMT1.2* and *IbAMT1.4* do not contain any introns, and members in the *AMT2* subfamily contain more introns than members in the *AMT1* subfamily.

### 2.8. Synteny Analysis

Gene family expansion and genomic evolution mechanisms depend on the gene duplication events that occurred during plant evolution [33,34]. We looked for duplication events and found three pairs of tandem *AMT* duplicated genes in the sweet potato chromosome (Figure 6), suggesting that some *IbAMT* genes may have been generated via gene duplication, and the tandem duplication played a significant role in the expansion of the *IbAMT* family. Moreover, the *IbAMT* gene family was subjected to strong purification selection pressure during evolution, because the Ka/Ks values of the syntenic gene pairs in this study were less than one (Appendix A).

Synteny analysis is a critical analytical strategy in comparative genomics that plays an essential role in assessing the molecular evolutionary relationships between species [35]. To better understand the phylogenetic relationships among the *AMT* family in plants, comparative syntenic maps of associations among sweet potato *AMT* and those found in two wild sweet potato species (*Ipomoea trifida* and *Ipomoea triloba*), cassava, tomato, *Arabidopsis thaliana*, and rice were constructed (Figure 7). Among all the *IbAMT* genes, 10 *IbAMT* genes (*IbAMT1.1*, *IbAMT1.2*, *IbAMT1.3*, *IbAMT1.4*, *IbAMT1.5*, *IbAMT2.1*, *IbAMT2.3*, *IbAMT2.4*, *IbAMT2.5*, *IbAMT2.7*) displayed a syntenic relationship with other crops.

### 2.9. Analysis of Cis-Regulatory Elements

Characterization of the promoter regions of genes is essential for understanding potential transcriptional regulatory mechanisms. PlantCare was used to analyze the 2000 bp upstream sequence of the starting point of *IbAMT* translation to detect cis elements. A total of 39 types of cis-regulatory elements (CREs) were detected in the promotor regions (Figure 8). According to the biological function of the tested elements, they can be divided into four categories, including light-responsive elements, hormone-responsive elements, stress-related elements, plant-development-related elements (Appendix A).

### 2.10. qRT-PCR Expression Analysis

Because *AMT1* play a more important role in NH_4_^+^ uptake than *AMT2* [25,26], we selected all *AMT1s* to design primers to determine their expressions. To understand the response of *IbAMT1* genes to NH_4_^+^ deficiency, the expressions of *IbAMT1* genes were analyzed in different tissues under NN using qRT-PCR. As shown in Figure 9, *IbAMT1.1* displayed an overall lower expression level than the other genes, and *IbAMT1.2* and *IbAMT1.4* were expressed to higher levels in the shoots than in the storage roots. Notably, the expressions of *IbAMT1.3* and *IbAMT1.5* in the storage roots were significantly higher than those in the leaves, petioles, and stems (*p* < 0.05).

Figure 10 shows that the expression level of *IbAMT1* genes was the highest in the leaves in the MN treatment, and the difference was significant (*p* < 0.05), except for *IbAMT1.1*. In the petioles, the expression level of *IbAMT1* genes of MN was significantly higher (*p* < 0.05) than that in NN. The expression level of *IbAMT1* genes was the highest in stems in MN, and the difference was significant (*p* < 0.05), except for *IbAMT1.3*. In the storage roots, the expression level of *IbAMT1* genes was the highest in MN, and the difference was significant (*p* < 0.05). The expression level of the *IbAMT1.5* gene was the highest, followed by *IbAMT1.3*.

### 2.11. Relationship between N Content and IbAMT1 Gene Expression

As shown in Figure 11, the expression of *IbAMT1* genes positively correlated with the N content in storage roots (*r* > 0). Among them, the expression of *IbAMT1.3* had a significant correlation (*r* = 0.97 *) with the N content, and the expression of *IbAMT1.5* had an extremely significant correlation (*r* = 1.00 ***) with the N content.

## 3. Discussion

### 3.1. Effect of N Uptake and Utilization on Growth, Development, and Yield

Nitrogen (N) is closely related to sweet potato yield [36]. Applying an appropriate amount of N is beneficial for increasing the number of storage roots, storage root weight, and yield of sweet potato [4,11]. The results of this study showed that the storage root number per plant, storage root weight, storage root weight per plant at canopy closure and harvest, and yield increased first and then decreased with increasing N application, which are findings similar to those reported in previous studies [7,37]. Meanwhile, compared with NN, the yield, storage root number per plant, and storage root weight in the MN treatment were significantly higher (*p* < 0.05). Additionally, MN plants had significantly increased leaf, petiole, and stem weight (*p* < 0.05) compared with NN plants at canopy closure and harvest. The above results showed that MN coordinated the development of N sources and sinks in sweet potato, which is similar to the results of previous studies on sweet potato [5,9]. Thus, the optimum N application rate is 120 kg ha^−1^ (MN) in this study.

N is the key nutrient element that regulates the growth and development of sweet potato [7], and N uptake is mainly concentrated within 60 DAP in sweet potato [38]. The results of present study support this conclusion: the N content in plant at canopy closure (50 DAP) was much higher than that at harvest. In this study, the N content in the leaf, petiole, stem, and storage root and N uptake efficiency increased at first and then decreased with the increase in N application, being highest in the MN treatment, and MN was significantly (*p* < 0.05) different from NN at 50 DAP. At harvest, the storage roots’ N content and NFCR increased first and then decreased with the increase in N application and were the highest in the MN treatment, which was significantly (*p* < 0.05) different from the NN treatment. The N contents in the leaf, petiole, and stem increased with the increase in N application. results are similar to those in previous studies on maize, wheat, and sweet potato [3,39,40,41]. Moreover, N use efficiency decreased with the increase in N application rate, mainly because the increase in the N content in shoots was higher than that in the storage roots with the increase in N application. This finding is consistent with those of previous studies by Du et al. [10,42]. To summarize, the main reasons for the increase in the number of storage roots per plant and storage root weight and yield in the MN plants may be the increase in the N content in the storage roots and NFCR. Previous studies in sweet potato and wheat support this conclusion [3,43].

### 3.2. Relationship between AMT Gene Family and N Uptake and Utilization

Sweet potato prefer ammonium over nitrate [44,45], and the ammonium transporter (AMT) plays crucial roles in ammonium transport [18]. The *AMT* gene family has been identified in many plants such as *Arabidopsis*, tomato, rice, and cassava [23,27,28,46,47] but not sweet potato. We performed a genome-wide analysis to identify 12 *AMT* family genes that belong to two subfamilies in sweet potato. The phylogenetic analyses of the AMT proteins among sweet potato, *Arabidopsis*, rice, tomato, cassava, soybean, and maize AMT proteins indicated that the AMT of the studied species can be divided into three groups according to homology [48]. Furthermore, *IbAMT1.3*, *AtAMT1.1*, and *SlAMT1.1* are homologous, and *IbAMT1.5* shows a close relationship with *AtAMT1.2* and *SlAMT1.2*. Phylogenetic and synteny analyses suggest that *IbAMT* genes were conserved during evolution. The differential expression of the *AMT* genes in the different plant tissues has been reported in many species. This study shows that *IbAMT1.3* and *IbAMT1.5* are mainly expressed in the roots, and their homologous genes *AMT1.1* and *AMT1.2* from tomato and *Arabidopsis* are also expressed in the roots [25,26,27]. Gansel suggested that NH_4_^+^ uptake is predominantly regulated by the local N status of the roots [49]. The upregulation of *AMT* genes can improve the ability of plant roots to uptake NH_4_^+^ [28,29,30]. In this study, the expressions of *IbAMT1.3* and *IbAMT1.5* in the roots increased first and then decreased with the increase in the N application at 50 DAP, which is similar to the findings of a previous study [19]. The expressions of *IbAMT1.3* and *IbAMT1.5* in the roots was the highest in the MN treatment, which was significantly higher than that in the NN treatment, indicating that the N uptake capacity of the roots of sweet potato was the highest in the MN treatment. Furthermore, there was a significant positive correlation between the expressions of *IbAMT1.3* and *IbAMT1.5* in the roots and the content of N in the storage roots, suggesting that *IbAMT1.3* and *IbAMT1.5* are related to N uptake. In this study, two candidate *IbAMT1* genes (*IbAMT1.3* and *IbAMT1.5*) for efficient N utilization in sweet potato were identified, but their molecular mechanism remains to be verified.

## 4. Materials and Methods

### 4.1. Experimental Site and Plant Material

Field experiments were conducted using sweet potato ‘Pushu 32’ (*Ipomoea batatas* (L.) Lam.) from October 2021 to February 2022 at the Agricultural Base of Hainan University (20°06′ N, 110°33′ E) and from October 2022 to February 2023 at Sanya Nanfan Research Institute of Hainan University (18°30′ N, 109°60′ E), respectively. In Table 6, climate data for the two growth seasons are listed, which were provided by Hefeng weather (https://www.qweather.com/ (accessed on 5 September 2023)); the details are shown in Table 1. The soil type of both fields was sandy loam. In Haikou, the organic matter concentration in the 0–20 cm soil layer of the field was 1.4%, and the total and available N, P, and K concentrations were 87.65, 16.00, and 77.80 mg kg^−1^ dry soil, respectively. In Sanya, the organic matter concentration was 1.7%, and the total and available N, P, and K concentrations were 83.67, 18.83, and 76.56 mg kg^−1^ dry soil, respectively.

### 4.2. Field Experiments

For the field experiments, we adopted a random blocks design with three replications. The used fertilizers were CH_4_N_2_O (urea, 46%), K_2_O (potassium sulfate, 50%), and P_2_O_5_ (calcium superphosphate, 16%), which were provided by Sinofert Holdings Limited (Beijing, China). Four different N fertilizer applications treatments were used: 0 kg ha^−1^ (NN), 60 kg ha^−1^ (LN), 120 kg ha^−1^ (MN), and 180 kg ha^−1^ (HN). Potassium fertilizer at a level of 240 kg K ha^−1^ and phosphate fertilizer at a level of 240 kg P ha^−1^ were applied in all treatments. The four treatment groups, each in quadruplicate, were allocated to different subplots. Each subplot had an area of 16 m^2^, with a row spacing of 0.8 m. The slips were spaced at 0.20 m and were planted at a depth of approximately 0.10 m in soil beds.

### 4.3. Plant Sampling and Analysis

The fresh samples were collected t 50 days after planting (DAP). The leaves, petioles, stems, and roots of each plant were mixed and then cut into 1 cm pieces, separately, then rapidly frozen with liquid N and stored at −80 °C for later enzymatic activity measurements and qRT-PCR. At canopy closure (50 DAP) and harvest (150 DAP) in the field, we selected five plants to investigate the fresh weight of the leaves, petioles, stems, and storage roots, and number of storage roots in sweet potato. Then, the plants were sterilized at 105 °C, dried at 60 °C, and the dry weight was recorded. Last, the dried plants were ground to powder and stored in a desiccator prior to quantification of N content. At harvest (150 DAP), roots that were greater than 1.0 cm in diameter were selected as storage roots. The number of storage roots per plant was counted, and the fresh weight of storage root was weighed to calculate the yield.

### 4.4. Total N Content

A Dumas N analyzer was used to determine the N content in different parts of sweet potato during the canopy closure and harvest. The Dumas N Analyzer uses the Primacs SN100 Dumas N Analyzer from SKALAR (Breda, The Netherlands).

### 4.5. Calculations

The indicators were calculated as follows [3,50,51,52,53]:N accumulation amount (g) = Plant dry matter accumulation × Plant N content(1)
N uptake efficiency (kg kg^−1^) = Plant N accumulation/Soil available N(2)
N use efficiency (kg kg^−1^) = Dry matter quality of storage roots/Plant N accumulation(3)
N fertilization contribution rates (NFCR) = (Yield in N application area − Yield in non-N application area)/Yield in N application area(4)

### 4.6. Sweet Potato AMT Family Genes’ Identification and Analysis

A hidden Markov model (HMM) describing the ammonium transporter domain (PF00909) was downloaded from the Pfam database (http://pfam.xfam.org/ (accessed on 25 April 2022)). This model was used to identify AMT proteins from the sweet potato genome (https://sweetpotao.com/ (accessed on 25 April 2022)) using a HMMER search program with an E-value cut of 1 × 10^−5^. All potential proteins were confirmed using Pfam, CDD (https://www.ncbi.nlm.nih.gov/cdd/ (accessed on 12 August 2022)), and SMART (http://smart.embl.de/smart/batch.pl (accessed on 12 August 2022)). ExPASy-ProtParam (http://web.expasy.org/protparam/ (accessed on 13 August 2022)) was used to analyze the physicochemical properties, the protein’s molecular weight (MW), and theoretical isoelectric point (pI) of the identified AMT proteins. WoLF PSORT Prediction (https://wolfpsort.hgc.jp/ (accessed on 2 November 2023)) was used to analyze the subcellular localization. The chromosome gene density and *IbAMT* locations on chromosomes were visualized using TBtools v1.098669 [54].

### 4.7. Phylogenetic Analysis

AMT proteins from *Arabidopsis*, rice (*Oryza sativa*), soybean (*Glycine max*) [55], maize (*Zea mays*) [56], cassava (*Manihot esculenta* Crantz) [23,28], and tomato (*Solanum lycopersicum*) [27] were downloaded from the TAIR (http://www.arabidopsis.org/ (accessed on 25 April 2022)), NCBI (https://www.ncbi.nlm.nih.gov/ (accessed on 25 April 2022)), and Phytozome (https://phytozome-next.jgi.doe.gov/ (accessed on 25 April 2022)) databases. Sequence similarity analysis was performed using the CLUSTALW website (https://www.genome.jp/tools-bin/clustalw (accessed on 13 August 2022)). The amino acid sequences of the seven species of AMT proteins were aligned using ClustalW [57]. Using the maximum likelihood (ML) method with 1000 bootstrap replications, a phylogenetic tree was generated with MEGA 11 (Arizona State University, Tempe, AZ, USA) software and clarified using the EvolView online tool (http://www.evolgenius.info/evolview (accessed on 14 May 2023))).

### 4.8. Conserved Motifs, Gene Structures, and Chromosomal Distribution Analysis

The amino acid sequences of the IbAMT proteins were aligned using ClustalW. Using the maximum-likelihood (ML) method with 1000 bootstrap replications, a phylogenetic tree was generated with MEGA 11 (Arizona State University, Tempe, AZ, USA) software. A conserved motif search was performed on IbAMT proteins using Multi Em for Motif Elicitation (http://meme-suite.org/tools/meme (accessed on 13 August 2022)) using the default parameters. The maximum number of motifs was set to 5 (width range = 1–100 motifs). Exon/intron information of *AMT* genes, including mRNA, CDS, and untranslated region (UTR) locations, were extracted from the sweet potato genome in a general feature format (GFF) file using a Biolinux system. Gene structure was visualized using TBtools v1.098669 [54].

### 4.9. Ka/Ks Analyses and Gene Collinearity

The nucleotide substitution parameters Ks (synonymous) and Ka (nonsynonymous) of the duplicated genes were assessed using TBtools v1.098669 [54], and then the Ka/Ks ratio was calculated [23]. The gene duplication events of sweet potato *AMT* and collinearity analysis with six other species were assessed and performed, respectively, according to the method of Yao et al. [58].

### 4.10. Cis-Regulatory Elements

Promoter sequences were extracted from the sweet potato genome database. The cis-regulatory elements were scanned into PlantCARE (http://bioinformatics.psb.ugent.be/webtools/plantcare/html/ (accessed on 24 August 2022)) and visualized using TBtools v1.098669 [54].

### 4.11. qRT-PCR Analysis

Total RNA was extracted using an RNA extraction kit according to the manufacturer’s protocol (Tiangen, Beijing, China, DP437). One microgram of RNA was reverse-transcribed to cDNA using HiScript II Q RT SuperMix for quantitative real-time PCR (qRT-PCR) (Vazyme, Nanjing, China, R223). To detect the transcript levels of the *IbAMT* genes in response to low N, quantitative qRT-PCR was performed with gene-specific primers using TB Green Premix Ex Taq II (Tli RNaseH Plus) (Takara, Tokyo, Japan, RR820A) on a qRT-PCR machine (qTOWER3G, Jena, Germany) for the qRT-PCR reaction. The housekeeping gene Actin was used as an internal control [59]. The relative expressions were calculated using the 2^−∆∆CT^ method. Three independent biological replicates were conducted. The primers used in this experiment are shown in Table 7.

### 4.12. Statistical Analysis

All data were statistically analyzed using Microsoft Excel 2003 (Microsoft, Redmond, WD, USA) to calculate means. The analytics software program SPSS 19.0 (IBM, Armonk, NY, USA) was used to conduct one-factor ANOVA, two-way ANOVA, and Duncan’s multiple range test.

Graphs in this study were visualized using GraphPad Prism 8.0.2.263 (GraphPad Software Inc., San Diego, CA, USA, www.graphpad.com (accessed on 24 August 2022)), TBtools v1.098669 [54], and R 4.0.2 [60].

## 5. Conclusions

This study explored the regulatory impacts of nitrogen (N) levels on N absorption and utilization, as well as the growth, development, and yield of sweet potato in a field experiment, identifying an optimum N application rate of 120 kg ha⁻^1^. Utilizing bioinformatics analysis and quantitative reverse-transcription PCR (qRT-PCR), two candidate *IbAMT1* genes (*IbAMT1.3* and *IbAMT1.5*) related to N uptake and accumulation in sweet potato were identified, establishing a basis for subsequent investigations into the molecular mechanism of efficient N utilization in sweet potato.

## Figures and Tables

**Figure 1 ijms-24-17424-f001:**
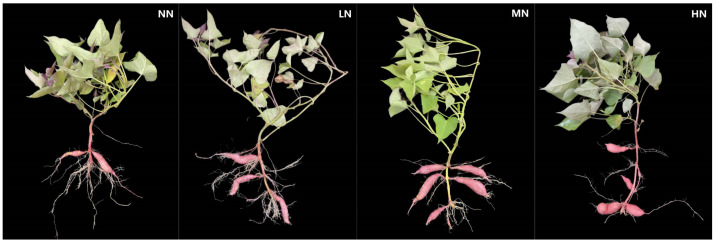
Growth of sweet potato at canopy closure. NN, no N; LN, 60 kg ha^−1^ N; MN, 120 kg ha^−1^ N; HN, 180 kg ha^−1^ N.

**Figure 2 ijms-24-17424-f002:**
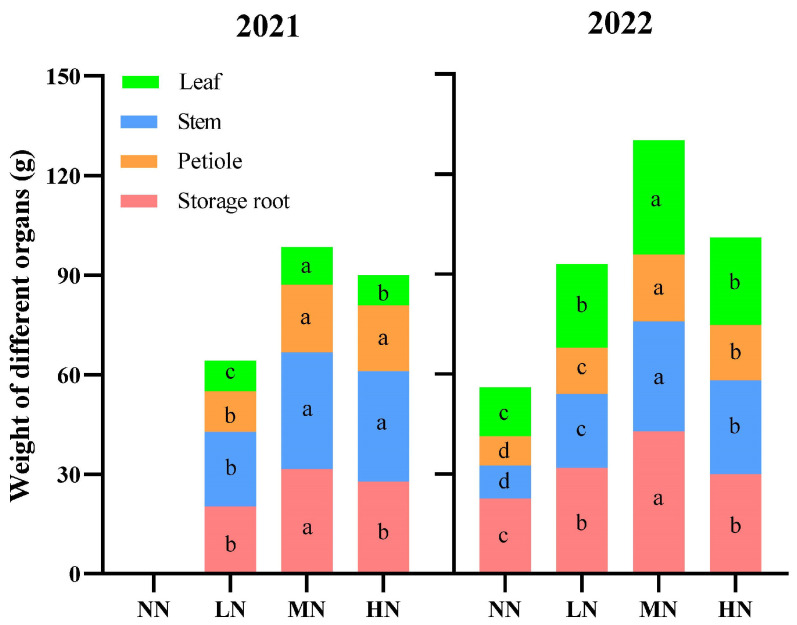
Weight of different organs of sweet potato at canopy closure. NN, no N; LN, 60 kg ha^−1^ N; MN, 120 kg ha^−1^ N; HN, 180 kg ha^−1^ N. Values followed by different letters within a group are significantly different among organs (*p* < 0.05).

**Figure 3 ijms-24-17424-f003:**
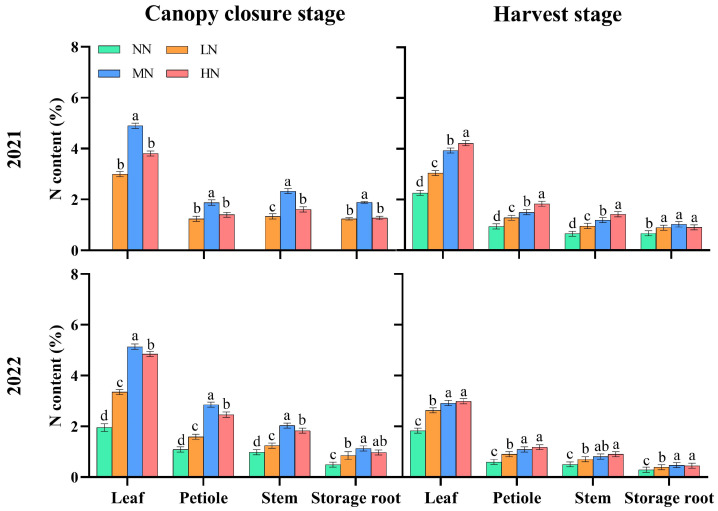
N content in different organs of sweet potato at canopy closure and harvest. NN, no N; LN, 60 kg ha^−1^ N; MN, 120 kg ha^−1^ N; HN, 180 kg ha^−1^ N. Values followed by different lowercase letters within a group are significantly different among nitrogen treatments (*p* < 0.05).

**Figure 4 ijms-24-17424-f004:**
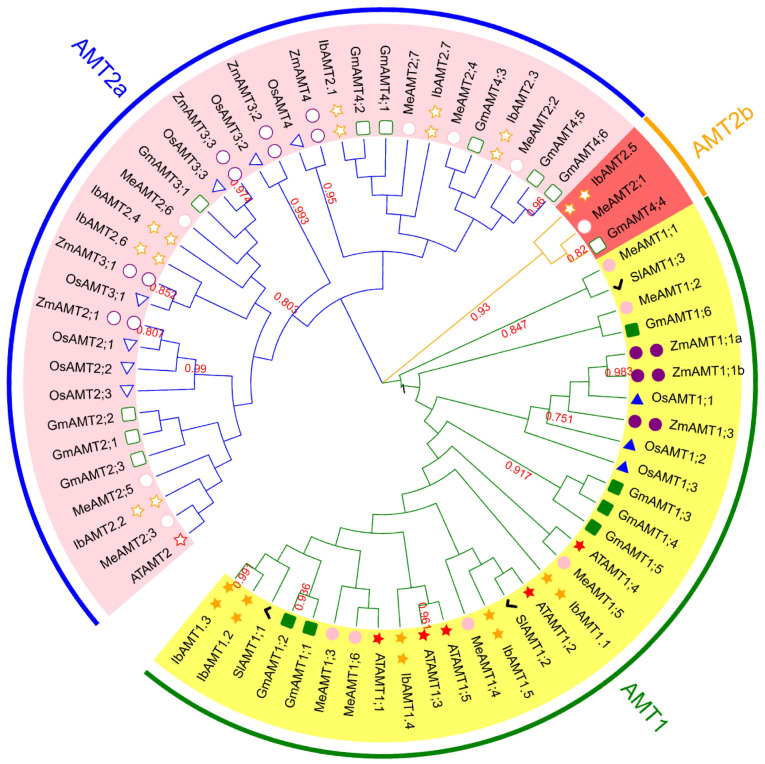
Phylogenetic tree for AMT proteins, constructed in MEGAX. The four primary subgroups were named Groups 1–3; different colors represent different subgroups. Pink circles, red stars, orange stars, green squares, blue triangles, purple circles, and black checks represent *MeAMTs*, *AtAMT*, *IbAMTs*, *GmAMTs*, *OsAMTs*, *ZmAMTs*, and *SlAMTs*; solid graphics represent *AMT1*; hollow graphics represent *AMT2*, respectively.

**Figure 5 ijms-24-17424-f005:**
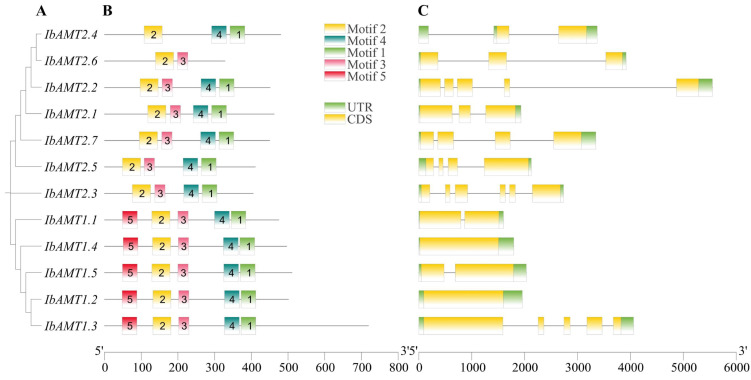
Phylogenetic relationships, conserved protein motifs, and gene structures of the 12 IbAMT proteins and genes. (**A**) The phylogenetic tree was constructed based on the full-length sequences of sweet potato AMT proteins using MEGA 7.0.26. (**B**) Motif distribution of the AMT proteins. The conserved motifs of AMT proteins were determined using MEME (http://meme-suite.org/tools/meme (accessed on 13 August 2022)) and visualized using TBtools v1.098669. The motifs, numbered 1–5, are displayed in different colored boxes. (**C**) *IbAMT* genes’ exon–intron structures. Black lines indicate introns, and yellow boxes represent exons.

**Figure 6 ijms-24-17424-f006:**
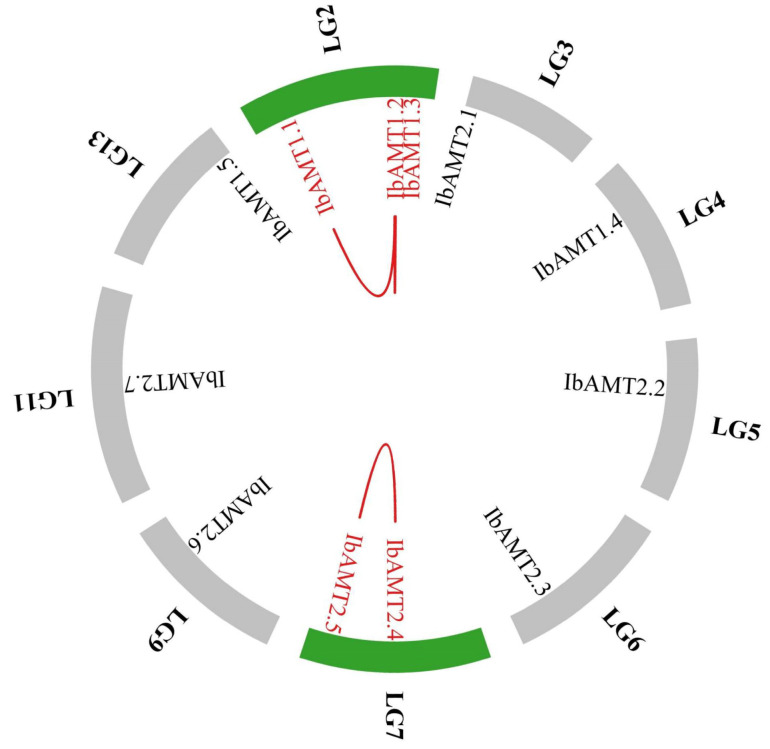
Schematic representations of the chromosomal distribution and interchromosomal relationships among sweet potato *AMT* genes. Chromosomes are represented in different colors. Red lines between *IbAMT* genes represent segmental duplication events that occurred in the sweet potato *AMT* gene family.

**Figure 7 ijms-24-17424-f007:**
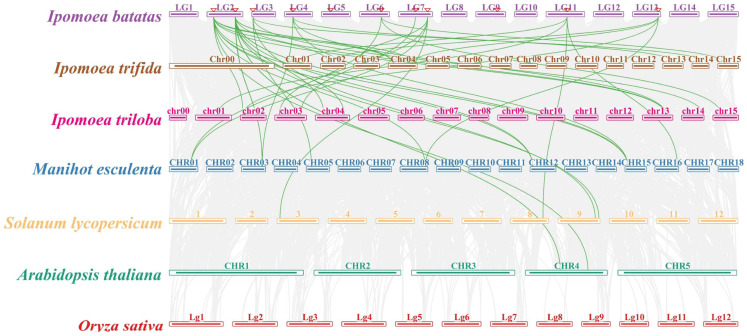
Synteny analyses of *AMT* genes between sweet potato and other representative plant species (*Arabidopsis*, cassava, rice, tomato, *Ipomoea trifida*, and *Ipomoea triloba*). Gray lines indicate significantly collinear blocks within and among plant genomes. Green lines indicate significantly collinear blocks within and among plant genomes between *AMT* genes. The red triangle represents the position of *IbAMT* genes on the chromosome of sweet potato.

**Figure 8 ijms-24-17424-f008:**
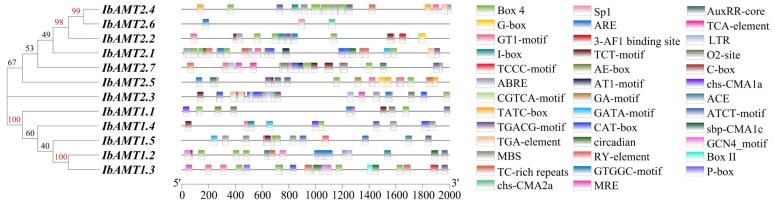
The cis-element analysis of *IbAMT* promoters.

**Figure 9 ijms-24-17424-f009:**
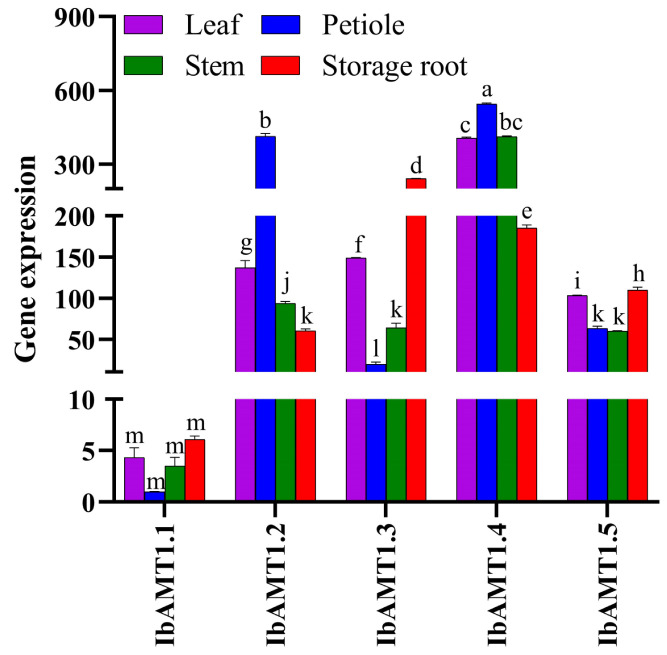
Expression profile of *IbAMT1* genes in different organs of sweet potato. Values followed by different lowercase letters within a group are significantly different among nitrogen treatments (*p* < 0.05).

**Figure 10 ijms-24-17424-f010:**
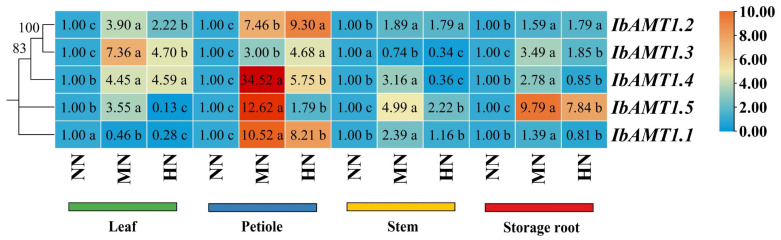
Expression of *IbAMT1* genes in storage root of sweet potato. NN, no N; MN, 120 kg ha^−1^ N; HN, 180 kg ha^−1^ N. Values followed by different lowercase letters within a group are significantly different among nitrogen treatments (*p* < 0.05).

**Figure 11 ijms-24-17424-f011:**
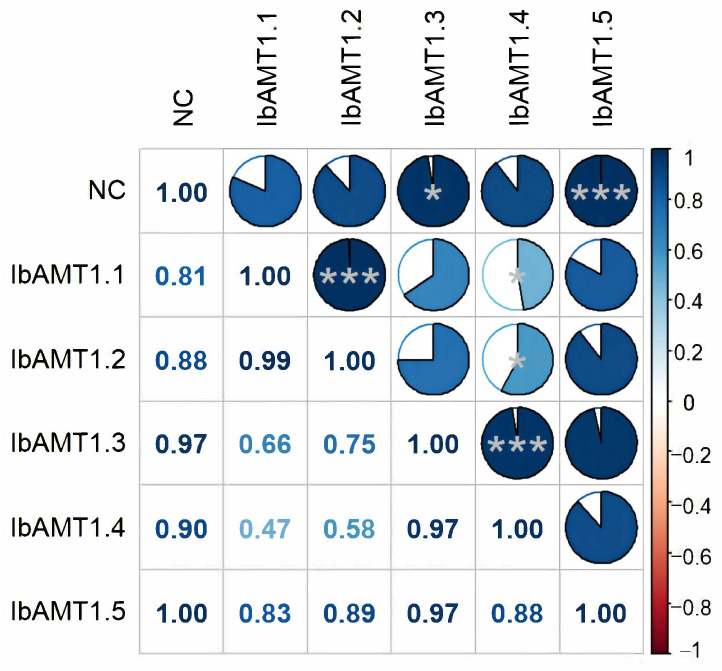
Correlation analysis between N content (NC) and *IbAMT1* gene expression in storage root of sweet potato. Blue represents positive correlation, and red represents negative correlation, where the darker the color, the stronger the correlation. * *p* < 0.05, *** *p* < 0.001.

**Table 1 ijms-24-17424-t001:** Yield components and yield.

Year	Treatment	Storage Root Number per Plant	Storage Root Weight (g)	Yield (kg ha^−1^)
2021	NN	3.00 ± 0.53 b	68.07 ± 2.87 b	10,035.58 ± 1717.58 c
	LN	3.50 ± 0.53 ab	87.81 ± 4.76 a	15,335.75 ± 2230.94 ab
	MN	3.88 ± 0.64 a	90.00 ± 6.08 a	17,356.83 ± 2501.77 a
	HN	3.25 ± 0.46 b	86.09 ± 2.85 a	13,975.53 ± 1901.70 b
2022	NN	2.75 ± 0.46 b	73.43 ± 4.18 b	10,108.88 ± 1867.65 c
	LN	3.25 ± 0.71 ab	84.83 ± 7.11 a	13,653.75 ± 2601.93 ab
	MN	3.50 ± 0.76 a	86.95 ± 4.69 a	15,179.02 ± 3123.49 a
	HN	2.88 ± 0.64 ab	81.99 ± 5.90 a	11,854.79 ± 3036.84 bc
ANOVA				
Year		1.56 *	22.80	34,895,725.64 *
Treatment		2.04 **	1017.34 **	110,543,895.31 **
Year × Treatment		0.02	77.42	4,467,957.23

Note: NN, no N; LN, 60 kg ha^−1^ N; MN, 120 kg ha^−1^ N; HN, 180 kg ha^−1^ N, two-way ANOVA, LSD. The values after the ± sign are the standard deviation; values followed by lowercase letters within a column are significantly different among N treatments (*p* < 0.05). * *p* < 0.05; ** *p* < 0.01.

**Table 2 ijms-24-17424-t002:** Agronomic characteristics of sweet potato at harvest.

Year	Treatment	Leaf Weight per Plant (g)	Petiole Weight per Plant (g)	Stem Weight per Plant (g)	Storage Root Weight per Plant (g)
2021	NN	41.87 ± 2.55 d	56.33 ± 3.31 d	74.43 ± 5.33 d	200.71 ± 34.35 c
	LN	76.66 ± 3.03 c	107.36 ± 5.16 c	123.61 ± 5.61 c	306.72 ± 44.62 ab
	MN	90.97 ± 5.38 b	126.88 ± 5.96 b	153.89 ± 3.73 b	347.14 ± 50.04 a
	HN	97.50 ± 2.09 a	150.07 ± 5.77 a	207.27 ± 5.64 a	279.51 ± 38.03 b
2022	NN	46.92 ± 1.69 c	43.23 ± 1.91 d	105.42 ± 5.69 d	202.18 ± 37.35 c
	LN	63.36 ± 5.03 b	74.53 ± 1.71 c	145.92 ± 4.49 c	273.08 ± 52.04 ab
	MN	88.81 ± 4.40 a	95.75 ± 2.50 b	185.98 ± 3.50 b	303.58 ± 62.47 a
	HN	90.78 ± 7.00 a	123.20 ± 5.22 a	204.81 ± 8.05 a	237.10 ± 34.54 bc
ANOVA					
Year		293.99 **	10,799.89 **	6880.54 **	13,958.29 *
Treatment		8260.67 **	21,558.12 **	39,354.79 **	44,217.56 **
Year × Treatment		238.19 **	320.08 **	1033.28 **	1787.18

Note: NN, no N; LN, 60 kg ha^−1^ N; MN, 120 kg ha^−1^ N; HN, 180 kg ha^−1^ N, two-way ANOVA, LSD. The values after the ± sign are the standard deviation; values followed by lowercase letters within a column are significantly different among N treatments (*p* < 0.05). * *p* < 0.05; ** *p* < 0.01.

**Table 3 ijms-24-17424-t003:** The storage root number per plant and weight at canopy closure.

Year	N Application Rate (kg ha^−1^)	Storage Root Number per Plant	Storage Root Weight per Plant (g)	Storage Root Weight per Plant (g)
2021	NN	-	-	-
	LN	3.50 ± 0.53 ab	9.22 ± 0.50 b	32.09 ± 3.90 b
	MN	4.13 ± 0.64 a	11.43 ± 1.65 a	46.36 ± 3.86 a
	HN	3.25 ± 0.71 b	9.12 ± 0.68 b	29.75 ± 6.99 b
2022	NN	2.63 ± 0.84 c	8.84 ± 2.02 b	22.60 ± 6.28 c
	LN	3.30 ± 0.64 ab	9.76 ± 1.64 b	31.79 ± 5.58 b
	MN	3.55 ± 0.50 a	12.07 ± 2.08 a	42.71 ± 9.14 a
	HN	2.98 ± 0.49 bc	10.20 ± 1.07 b	29.96 ± 2.74 b
ANOVA analysis				
Year		1.47	6.81	18.77
Treatment		2.27 **	23.70 **	971.80 **
Year × Treatment		0.16	0.33	17.53

Note: NN, no N; LN, 60 kg ha^−1^ N; MN, 120 kg ha^−1^ N; HN, 180 kg ha^−1^ N, two-way ANOVA, LSD. The values after the ± sign are the standard deviation; values followed by lowercase letters within a column are significantly different among N treatments (*p* < 0.05). ** *p* < 0.01.

**Table 4 ijms-24-17424-t004:** N use efficiencies of sweet potato at canopy closure and harvest.

Period	Year	Treatment	N Uptake Efficiency (kg kg^−1^)	N Use Efficiency (kg kg^−1^)	N Fertilization Contribution Rates
Canopy Closure Period	2021	NN	-	-	-
		LN	19.49 ± 0.27 b	0.30 ± 0.01 a	-
		MN	25.42 ± 0.16 a	0.17 ± 0.01 b	-
		HN	9.54 ± 0.09 c	0.18 ± 0.01 b	-
	2022	NN	-	0.58 ± 0.05 a	-
		LN	22.72 ± 1.19 b	0.27 ± 0.02 b	-
		MN	27.66 ± 0.67 a	0.17 ± 0.01 c	-
		HN	11.86 ± 0.23 c	0.15 ± 0.01 c	-
Harvest Time	2021	NN	-	0.78 ± 0.09 a	-
		LN	107.04 ± 8.45 a	0.55 ± 0.05 b	0.35 ± 0.02 b
		MN	73.96 ± 3.86 b	0.45 ± 0.02 c	0.42 ± 0.01
		HN	44.01 ± 2.58 c	0.39 ± 0.02 c	0.28 ± 0.01
	2022	NN	-	1.17 ± 0.22 a	-
		LN	68.49 ± 8.43 a	0.77 ± 0.10 b	0.26 ± 0.01
		MN	47.29 ± 5.04 b	0.62 ± 0.07 bc	0.33 ± 0.02
		HN	32.72 ± 3.04 c	0.44 ± 0.05 c	0.15 ± 0.01
ANOVA					
Canopy Closure Period	Year		30.42 **	0	-
	Treatment		388.74 **	0.13 **	-
	Year × Treatment		0.45	0	-
Harvest Time	Year		2926.28 **	0.26 **	0.05 **
	Treatment		3672.63 **	0.35 **	0.04 **
	Year × Treatment		280.17 **	0.03	0.00 **

Note: Table 4 was calculated with original data from Appendix A by the formula in Section 4.5. Calculations. NN, no N; LN, 60 kg ha^−1^ N; MN, 120 kg ha^−1^ N; HN, 180 kg ha^−1^ N, two-way ANOVA, LSD. The values after the ± sign are standard deviation; values followed by lowercase letters within a column are significantly different among N treatments (*p* < 0.05). ** *p* < 0.01.

**Table 5 ijms-24-17424-t005:** Features of *AMT* family members in sweet potato. (^a^ Length (no. of amino acids) of the deduced polypeptide, ^b^ molecular weight of the deduced polypeptide, ^c^ isoelectric point of the deduced polypeptide).

Gene ID	Rename	Chromosome	+/−	From	To	Protein (aa) ^a^	MW (Da) ^b^	PI ^c^	Instability Index	Subcellular Localization
*g5252.t1*	*IbAMT1.1*	LG2	−	6,966,130	6,967,728	474	50,364.83	5.94	26.04	Plasma membrane
*g8186.t1*	*IbAMT1.2*	LG2	−	29,455,175	29,457,130	500	53,162.08	6.49	25.23	Plasma membrane
*g8219.t1*	*IbAMT1.3*	LG2	+	29,703,630	29,707,688	718	76,762.85	8.54	45.62	Plasma membrane
*g9593.t1*	*IbAMT2.1*	LG3	+	1,155,199	1,157,128	461	50,746.34	8.49	42.93	Plasma membrane
*g14003.t1*	*IbAMT1.4*	LG4	+	8,704,965	8,706,760	495	52,496.55	7.11	23.74	Cytosol
*g17995.t1*	*IbAMT2.2*	LG5	+	9,624,603	9,630,151	450	48,335.51	8.77	28.96	Plasma membrane
*g23921.t1*	*IbAMT2.3*	LG6	−	22,757,713	22,760,447	404	44,089.46	6.22	32.83	Plasma membrane
*g27616.t1*	*IbAMT2.4*	LG7	+	17,584,800	17,610,979	480	52,074.74	8.34	35.08	Plasma membrane
*g29440.t1*	*IbAMT2.5*	LG7	−	30,456,113	30,458,239	410	44,766.8	6.34	37.27	Plasma membrane
*g37079.t1*	*IbAMT2.6*	LG9	−	22,670,108	22,674,025	327	35,229.26	8.74	34.14	Plasma membrane
*g44418.t1*	*IbAMT2.7*	LG11	−	22,000,413	22,003,757	449	48,838.24	6.30	31.50	Plasma membrane
*g54923.t1*	*IbAMT1.5*	LG13	+	26,814,002	26,816,033	510	54,360.35	6.85	27.43	Plasma membrane

**Table 6 ijms-24-17424-t006:** Climate during the sweet potato growing season.

City	Year	Month	Total Rainfall (mm)	Maximum Temperature (°C)	Minimum Temperature (°C)	Average Temperature (°C)
Haikou	2021	10	505.30	32.00	18.00	24.90
		11	26.30	30.00	17.00	22.70
		12	57.90	25.00	14.00	19.00
	2022	1	16.70	26.00	14.00	19.70
		2	140.60	26.00	8.00	16.80
Sanya		10	45.00	31.00	20.00	25.70
		11	2.10	31.00	20.00	25.60
		12	0.20	29.00	15.00	21.70
	2023	1	3.00	28.00	14.00	21.30
		2	9.00	31.00	18.00	23.40

**Table 7 ijms-24-17424-t007:** Primers for qRT-PCR.

Gene Name	Forward Primer Sequence (5′-3′)	Reverse Primer Sequence (5′-3′)
*Actin*	TATGGTTGGGATGGGACAGAA	CGGTAAGAAGGACAGGGTGCT
*IbAMT1.1*	CAACGGCGTGGAAGACAAATTCG	GAAGACTAGGTAGGCGGAGAAGAGG
*IbAMT1.2*	GGAAGACGAAATGGCGGGTATGG	TGCGGGTTCAATCCTTCTCATTTGG
*IbAMT1.3*	CATATCCATACCGACGCCCATGTAC	CTCCTCCCTCCCATCTCTCATCAAG
*IbAMT1.4*	TGTCGGGGCATTGGAAAGTTACG	ATTAAGACCAGAGCCGCCACAAAC
*IbAMT1.5*	CGGAAGATGAGACCTGCGGAATG	GTGTGGGAGTATTGGACGGTTCG

## Data Availability

Data are available on request due to restrictions, e.g., privacy-based or ethical restrictions.

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
