# Peer review of "Genome-Wide Analysis of Sweet Potato Ammonium Transporter (AMT): Influence on Nitrogen Utilization, Storage Root Development and Yield"

_ijms, 2023, doi:10.3390/ijms242417424_

Round 1

Reviewer 1 Report

Comments and Suggestions for Authors

Manuscript Number: IJMS-2751207 20231124 Comments

Article Type: Article

Section/Category: Molecular Plant Sciences

Title: Genome-wide Analysis of Sweet Potato Ammonium Transporter (AMT)-Family Genes and Revealing IbAMT1.3 and IbAMT1.5 Maybe Influence Nitrogen Utilization, Storage Root Development and Yield

Authors: Ya-yi Meng , Ning Wang , Hai-yan Zhang , Ran Xu, Cheng-cheng Si *

The article is focused on nitrogen metabolism.

The major issues which need to be addressed are:

Q1: As stated, urea metabolism supplies ammonium to the plant.

       And that ammonium influences N storage.

  It is recommended that in experiments, other sources of ammonium should have been included to justify the results.

         Here, ammonium sulphate, ammonium chloride and ammonium nitrate and potassium nitrate should have been taken into consideration.

        These sources would have clearly indicated:

(i)                 Is ammonium the key player? and

(ii)              Is ammonium metabolism influenced by the presence of nitrate?

Hence, either the authors can conduct the suggested experiments.

              or     search published literature and discuss their data, accordingly.

Minor issue:

Title: Genome-wide Analysis of Sweet Potato Ammonium Transporter (AMT)-Family Genes and Revealing IbAMT1.3 and IbAMT1.5 Maybe Influence Nitrogen Utilization, Storage Root Development and Yield

Q2: “May be” can be deleted.

Q3: The introduction has some issues regarding English grammar.

Comments on the Quality of English Language

Introduction may be revised.

Author Response

(1) Q1: As stated, urea metabolism supplies ammonium to the plant. And that ammonium influences N storage. It is recommended that in experiments, other sources of ammonium should have been included to justify the results. Here, ammonium sulphate, ammonium chloride and ammonium nitrate and potassium nitrate should have been taken into consideration. These sources would have clearly indicated:

(i) Is ammonium the key player? And

(ii) Is ammonium metabolism influenced by the presence of nitrate?

Hence, either the authors can conduct the suggested experiments. Or search published literature and discuss their data, accordingly.

Answer: Thanks. We added ‘Both ammonium and nitrate are crucial for plant growth and yield, ammonium requires less energy to assimilate, therefore, ammonium nitrogen is preferentially absorbed [13], and considered as a superior nitrogen source [14].’ to explain that ammonium is the key player. Lines 55-57.

At the same time, we added information on the sources of ammonium used in the previously mentioned experiments, such as NH4Cl and (NH4)2SO4. Besides, we added ‘This suggests that increased NH4+ concentration significantly reduces the expression of OsAMT1.1. Compared with ammonium, nitrate can inhibit the absorption of ammonium salts, resulting in a decrease in the expression of OsAMT genes (OsAMT1;1, OsAMT1;2 and OsAMT1;3) under ammonium–nitrate mixed conditions [31]. Studies in Arabidopsis revealed that ammonium transporters, such as AtAMT1.1 AtAMT1.3 and AtAMT2, are transcriptionally regulated by urea [32]. Therefore, it can be concluded that urea has the ability to affect the activity of AMT genes and the up-regulation of these genes can enhance the plant roots’ ability to uptake NH4+.’ to explain if ammonium metabolism influenced by the presence of nitrate and why we chose urea. Lines 85-93.

Minor issue:

Title: Genome-wide Analysis of Sweet Potato Ammonium Transporter (AMT)-Family Genes and Revealing IbAMT1.3 and IbAMT1.5 Maybe Influence Nitrogen Utilization, Storage Root Development and Yield

(2) Q2: “May be” can be deleted.

Answer: Thanks. We revised the title as ‘Genome-wide Analysis of Sweet Potato Ammonium Trans-porter (AMT): Influence Nitrogen Utilization, Storage Root Development and Yield’.

(3) Q3: The introduction has some issues regarding English grammar. Comments on the Quality of English Language Introduction may be revised.

Answer: We have asked native English-speaking experts to help improve the language of the introduction.

Sweet potato [Ipomoea batatas (L.) Lam.] yield is determined by storage root development, and nitrogen (N) is the key nutrient to regulate storage root development [1]. Too high or too low N application is not conducive to storage root formation: the N deficiency which reduces the dry matter accumulation, is not conducive to storage root formation, and reduces the yield of sweet potato [2], although N use efficiency is higher [3]; appropriate N is beneficial to increase storage root number, improving dry matter accumulation in storage root and yield of sweet potato [3-5]; excessive N application leads to excessive growth of shoots [6], increased the plant N accumulation [7-9] and dry matter accumulation of whole plant, but delayed storage root formation and significantly reduced the N use efficiencies (NUE), and decreased yield of sweet potato [10-12]. Therefore, proper N application is an effective measure to promote storage root formation and increase the yield of sweet potato.

Both ammonium and nitrate are crucial for plant growth and yield, ammonium requires less energy to assimilate, therefore, ammonium nitrogen is preferentially absorbed [13], and considered as a superior nitrogen source [14]. Urea is the most commonly used N fertilizer because of its low cost, high N content, and rapid N release [15], it can be hydrolyzed into ammonium (NH4+) by urease [16]. The transportation of NH4+ by the roots from the soil is the start of ammonia assimilation and the foundational step for N utilization [17]. Ammonium transporter (AMT) exists widely in plants, a kind of plasma membrane protein that exclusively transports NH4+ [18]. The first ammonium transporter AtAMT1.1 was isolated from the model plant Arabidopsis through growth complementarity [19]. Subsequently, several AMT genes were identified in Arabidopsis, rice, tomato, and other plant species [20-22]. Plant AMT genes form a polygenic family divided into AMT1s and AMT2s subfamilies according to gene homology [23]. For example, there are 5 AMT1 genes and 1 AMT2 gene in Arabidopsis thaliana [24]. Among them, AtAMT1.2, AtAMT1.3, AtAMT1.5, and AtAMT2 were found to be mainly expressed in roots, AtAMT1.4 showed pollen-specific expression and AtAMT1.1 was expressed in roots, stems and leaves [25,26]. In rice, OsAMT1;2 and OsAMT1;3 were found to be mainly expressed in roots, while OsAMT1;1, OsAMT2;1, and OsAMT3;1 were expressed in both roots and shoots [19]. In tomatoes, SlAMT1-1 and SlAMT1-2 were found to be mainly expressed in roots and leaves, and SlAMT1-3 was expressed in seedlings and leaves [27]. AMT genes are differentially expressed. Besides, the expression of the AMT gene is affected by nitrogen levels and nitrogen forms. Studies in cassava showed that the expression of all MeAMT1 genes in roots was up-regulated under low NH4+ (NH4Cl) stress [28]. After the rape seedlings cultured in water for 10 days under N (NH4Cl)-suitable condition were transferred to N (NH4Cl)-deficient environment for 5 days, it was found that the expression of all AMT genes in roots was up-regulated [29]. In rice, the transcript levels of OsAMT1.1 in roots exhibit a several-fold decreased within 48 h when plants acclimated to 10 uM external NH4+ ((NH4)2SO4) for 3 weeks are transferred to 10 mM NH4+ ((NH4)2SO4) [30]. This suggests that increased NH4+ concentration significantly reduces the expression of OsAMT1.1. Compared with ammonium, nitrate can inhibit the absorption of ammonium salts, resulting in a decrease in the expression of OsAMT genes (OsAMT1;1, OsAMT1;2 and OsAMT1;3) under ammonium–nitrate mixed conditions [31]. Studies in Arabidopsis revealed that ammonium transporters, such as AtAMT1.1 AtAMT1.3 and AtAMT2, are transcriptionally regulated by urea [32]. Therefore, it can be concluded that urea has the ability to affect the activity of AMT genes and the up-regulation of these genes can enhance the plant roots’ ability to uptake NH4+. However, when compared to other crops, there is a lack of research on the AMT gene family in sweet potatoes and the function of IbAMT transporters in NH4+ absorption and utilization. Additionally, it remains unclear whether the expression of IbAMTs is related to sweet potato storage root development and yield under different nitrogen treatments.

To address this gap in knowledge, this study aims to identify the members of the AMT gene family in sweet potatoes and screen for key IbAMT genes that may regulate N utilization and impact storage root development and yield in sweet potatoes. The findings of this study will provide a theoretical basis for further investigation into the function and mechanism of IbAMT regulating N uptake and utilization in sweet potatoes.

Reviewer 2 Report

Comments and Suggestions for Authors

Comments

The paper titled "Genome-wide Analysis of Sweet Potato Ammonium Transporter (AMT)-Family Genes and Revealing IbAMT1.3 and IbAMT1.5 Maybe Influence Nitrogen Utilization, Storage Root Development and Yield" presents a comprehensive analysis of AMT genes in sweet potatoes and their potential impact on nitrogen utilization and yield. The study is well-structured and employs robust methods for gene identification, phylogenetic analysis, and motif composition.

The study conducted a comprehensive analysis of sweet potato AMT-family genes, shedding light on their role in nitrogen utilization, storage root development, and yield. The methods employed for gene identification, phylogenetic analysis, and motif composition were well-executed and sound. However, I have a few suggestions and concerns:

The abstract could benefit from a brief description of the sweet potato cultivar 'Pushu 32' and the rationale behind selecting it for field experiments. Additionally, providing information on the geographical location and conditions of the field experiments could add context to the study.

It would be helpful to provide more details on the specific experimental setup, such as the plot size, replication, and statistical methods used for data analysis. This information would enhance the reproducibility of the results.

While the results are intriguing, further discussion is needed to elucidate the potential mechanisms by which IbAMT1.3 and IbAMT1.5 influence nitrogen utilization, storage root development, and yield. Are there any proposed pathways or interacting genes that could be involved in these processes?

The study mentions a significant positive correlation between the expression of IbAMT1.3 and IbAMT1.5 with nitrogen content in storage roots. It would be beneficial to include the correlation coefficient (r-value) to quantify the strength of this relationship.

In the discussion section, it would be valuable to explore the potential molecular mechanisms by which IbAMT1.3 and IbAMT1.5 affect nitrogen utilization and storage root development. Are there any known pathways or regulators that these genes interact with? This would provide a deeper understanding of the biological processes involved.

The paper could benefit from a brief conclusion section summarizing the key findings and their implications. This would help readers quickly grasp the main takeaways from the study.

Overall, this research significantly contributes to our understanding of sweet potato genetics and nitrogen utilization. Addressing the above points would further enhance the clarity and impact of the paper.

Overall, this research is valuable and contributes to our understanding of the genetic factors influencing nitrogen utilization in sweet potatoes. Addressing the above points would further enhance the clarity and impact of the study.

Comments on the Quality of English Language

The manuscript need to revised as Major Revision 

Author Response

(1) The abstract could benefit from a brief description of the sweet potato cultivar 'Pushu 32' and the rationale behind selecting it for field experiments. Additionally, providing information on the geographical location and conditions of the field experiments could add context to the study.

Answer: Thanks! We changed as ‘In the present study, the sweet potato cultivar ‘Pushu 32’ which planted in a large area in China was used in field experiments at Agricultural Base of Hainan University (20° 06'N, 110° 33'E) in 2021, and at Sanya Nanfan Research Institute of Hainan University (18° 30'N, 109° 60'E) in 2022. Four N levels were tested: 0, 60, 120 and 180 kg ha-1.’

(2) It would be helpful to provide more details on the specific experimental setup, such as the plot size, replication, and statistical methods used for data analysis. This information would enhance the reproducibility of the results.

Answer: Thanks. These contents are expounded in our materials and methods. ‘The field experiments adopted random blocks design with three replications. The used fertilizers were CH4N2O (urea, 46%), K2O (potassium sulfate, 50%) and P2O5 (calcium superphosphate, 16%) provided by Sinofert Holdings Limited (Beijing, China). Four different N fertilizer applications treatments were used: 0 kg ha-1 (NN), 60 kg ha-1 (LN), 120 kg ha-1 (MN), 180 kg ha-1 (HN). Potassium fertilizer at a level of 240 kg K ha-1 and phosphate fertilizer at a level of 240 kg P ha-1 were applied in all treatments. The Four treatment groups, each in quadruplicate, allocated to different subplots. Each subplot had an area of 16 m2, with a row spacing of 0.8 m. The slips were spaced at 0.20 m and planted at a depth of approximately 0.10 m in soil beds.’

(3) While the results are intriguing, further discussion is needed to elucidate the potential mechanisms by which IbAMT1.3 and IbAMT1.5 influence nitrogen utilization, storage root development, and yield. Are there any proposed pathways or interacting genes that could be involved in these processes?

Answer: Unfortunately, the genes interacting with IbAMT1.3 and IbAMT1.5 are still unknown, but this is what we will continue to study later.

(4) The study mentions a significant positive correlation between the expression of IbAMT1.3 and IbAMT1.5 with nitrogen content in storage roots. It would be beneficial to include the correlation coefficient (r-value) to quantify the strength of this relationship.

Answer: The number on the left of figure 11 represents the r-value.

(5) In the discussion section, it would be valuable to explore the potential molecular mechanisms by which IbAMT1.3 and IbAMT1.5 affect nitrogen utilization and storage root development. Are there any known pathways or regulators that these genes interact with? This would provide a deeper understanding of the biological processes involved.

Answer: Unfortunately, the pathways and regulators interacting with IbAMT1.3 and IbAMT1.5 are still unknown, but this is what we will continue to study later.

(6) The paper could benefit from a brief conclusion section summarizing the key findings and their implications. This would help readers quickly grasp the main takeaways from the study.

Answer: Thank you for your objective comments. We revised conclusion as ‘This study explored the regulatory impacts of nitrogen (N) levels on N absorption and utilization, as well as the growth, development, and yield of sweet potato in a field experiment, identifying an optimum N application rate of 120 kg ha⁻¹. Utilizing bioinformatics analysis and quantitative reverse transcription PCR (qRT-PCR), two candidate IbAMT1 genes (IbAMT1.3 and IbAMT1.5) related to N uptake and accumulation in sweet potato were identified, establishing a basis for subsequent investigations into the molecular mechanism of efficient N utilization in sweet potato.’ Lines 475-481.

(7) Overall, this research significantly contributes to our understanding of sweet potato genetics and nitrogen utilization. Addressing the above points would further enhance the clarity and impact of the paper. Overall, this research is valuable and contributes to our understanding of the genetic factors influencing nitrogen utilization in sweet potatoes. Addressing the above points would further enhance the clarity and impact of the study.

Answer: Thank you for your objective comments.

Reviewer 3 Report

Comments and Suggestions for Authors

Dear Authors,

The manuscript titled "Genome-wide Analysis of Sweet Potato Ammonium Transporter (AMT)-Family Genes and Revealing IbAMT1.3 and IbAMT1.5 Maybe Influence Nitrogen Utilization, Storage Root Development and Yield" authored by Ya-yi Meng, Ning Wang, Hai-yan Zhang, Ran Xu, Cheng-cheng Si presents a comprehensive exploration into the AMT gene family in sweet potatoes and screen the key IbAMT genes and their potential impact on nitrogen utilization, storage root development, and yield. The study utilized the sweet potato cultivar ‘Pushu 32’ in field experiments conducted in 2021 and 2022, examining various nitrogen levels to assess its effects on gene expression and plant development.

The chapter Introduction is rather weak in providing a robust background for the presented results. However, the chapter Materials and Methods, detailing the methodology is well-written, offering a clear understanding of the experimental approach employed. The Discussion section is thoughtfully constructed, effectively juxtaposing the authors' findings against existing literature, substantiated by an extensive list of publications. The authors in the Discussion section, skillfully integrate their research outcomes with prior studies, elucidating the implications of their findings in the broader context of sweet potato physiology and genetics.

Despite the manuscript's strengths, there are certain shortcomings that need addressing:

1) I propose shortening the title of the manuscript, it could read, for example: "Genome-wide Analysis of Sweet Potato Ammonium Transporter (AMT): Influence Nitrogen Utilization, Storage Root Development and Yield".

2) Abstract - both the purpose of the research and the research hypothesis are too poorly outlined here. Please specify the application significance of the research results.

3) Keywords - please do not use words that have already appeared in the title of the manuscript. Furthermore, keywords should be arranged in alphabetical order.

4) Introduction - lacks the depth necessary to sufficiently contextualize the research, making it challenging for readers to grasp the significance of the subsequent findings. Lines 75-79 - please outline your research hypothesis better here.

5) Tables 1, 2, 3 and 4 - explain what the values after the +/- sign mean; instead of "Year * Treatment" give "Year x Treatment".

6) Figure 1 - explain the symbols NN, LN, MN and HN under the figure.

7) Figure 2 - on the Y axis, instead of "weight" put "Weight of ... " (what?). Explain what the letters on the bars and the symbols on the x-axis mean.

8) Table 3 - in the second heading, replace "Kg" with "kg".

9) Lines 114 and 136 - remove "*—p<0.05" - this case is not in the table.

10) Figure 3 - on the Y axis, replace "N" with "N content". Explain the designations NN, LN, MN and HN under the figure. Explain what the letters above the bars mean.

11) Figures 9 and 10 - explain what the letters above the bars mean.

12) Line 247 - remove "**—p<0.01" - this case is not in the figure.

13) Line 306 - add "...and Plant Material".

14) Line 307 - please add the scientific name of the species.

In conclusion, the manuscript presents valuable insights into the AMT gene family's role in the case of N utilization and affects storage root development and yield in sweet potatoes.

To sum up, I believe that the Dear Editors of the IJMS journal should consider publishing this manuscript.

Author Response

(1) The manuscript titled "Genome-wide Analysis of Sweet Potato Ammonium Transporter (AMT)-Family Genes and Revealing IbAMT1.3 and IbAMT1.5 Maybe Influence Nitrogen Utilization, Storage Root Development and Yield" authored by Ya-yi Meng, Ning Wang, Hai-yan Zhang, Ran Xu, Cheng-cheng Si presents a comprehensive exploration into the AMT gene family in sweet potatoes and screen the key IbAMT genes and their potential impact on nitrogen utilization, storage root development, and yield. The study utilized the sweet potato cultivar ‘Pushu 32’ in field experiments conducted in 2021 and 2022, examining various nitrogen levels to assess its effects on gene expression and plant development. The chapter Introduction is rather weak in providing a robust background for the presented results. However, the chapter Materials and Methods, detailing the methodology is well-written, offering a clear understanding of the experimental approach employed. The Discussion section is thoughtfully constructed, effectively juxtaposing the authors' findings against existing literature, substantiated by an extensive list of publications. The authors in the Discussion section, skillfully integrate their research outcomes with prior studies, elucidating the implications of their findings in the broader context of sweet potato physiology and genetics.

Answer: Thank you for your objective comments. We have revised the introduction.

Sweet potato [Ipomoea batatas (L.) Lam.] yield is determined by storage root development, and nitrogen (N) is the key nutrient to regulate storage root development [1]. Too high or too low N application is not conducive to storage root formation: the N deficiency which reduces the dry matter accumulation, is not conducive to storage root formation, and reduces the yield of sweet potato [2], although N use efficiency is higher [3]; appropriate N is beneficial to increase storage root number, improving dry matter accumulation in storage root and yield of sweet potato [3-5]; excessive N application leads to excessive growth of shoots [6], increased the plant N accumulation [7-9] and dry matter accumulation of whole plant, but delayed storage root formation and significantly reduced the N use efficiencies (NUE), and decreased yield of sweet potato [10-12]. Therefore, proper N application is an effective measure to promote storage root formation and increase the yield of sweet potato.

Both ammonium and nitrate are crucial for plant growth and yield, ammonium requires less energy to assimilate, therefore, ammonium nitrogen is preferentially absorbed [13], and considered as a superior nitrogen source [14]. Urea is the most commonly used N fertilizer because of its low cost, high N content, and rapid N release [15], it can be hydrolyzed into ammonium (NH4+) by urease [16]. The transportation of NH4+ by the roots from the soil is the start of ammonia assimilation and the foundational step for N utilization [17]. Ammonium transporter (AMT) exists widely in plants, a kind of plasma membrane protein that exclusively transports NH4+ [18]. The first ammonium transporter AtAMT1.1 was isolated from the model plant Arabidopsis through growth complementarity [19]. Subsequently, several AMT genes were identified in Arabidopsis, rice, tomato, and other plant species [20-22]. Plant AMT genes form a polygenic family divided into AMT1s and AMT2s subfamilies according to gene homology [23]. For example, there are 5 AMT1 genes and 1 AMT2 gene in Arabidopsis thaliana [24]. Among them, AtAMT1.2, AtAMT1.3, AtAMT1.5, and AtAMT2 were found to be mainly expressed in roots, AtAMT1.4 showed pollen-specific expression and AtAMT1.1 was expressed in roots, stems and leaves [25,26]. In rice, OsAMT1;2 and OsAMT1;3 were found to be mainly expressed in roots, while OsAMT1;1, OsAMT2;1, and OsAMT3;1 were expressed in both roots and shoots [19]. In tomatoes, SlAMT1-1 and SlAMT1-2 were found to be mainly expressed in roots and leaves, and SlAMT1-3 was expressed in seedlings and leaves [27]. AMT genes are differentially expressed. Besides, the expression of the AMT gene is affected by nitrogen levels and nitrogen forms. Studies in cassava showed that the expression of all MeAMT1 genes in roots was up-regulated under low NH4+ (NH4Cl) stress [28]. After the rape seedlings cultured in water for 10 days under N (NH4Cl)-suitable condition were transferred to N (NH4Cl)-deficient environment for 5 days, it was found that the expression of all AMT genes in roots was up-regulated [29]. In rice, the transcript levels of OsAMT1.1 in roots exhibit a several-fold decreased within 48 h when plants acclimated to 10 uM external NH4+ ((NH4)2SO4) for 3 weeks are transferred to 10 mM NH4+ ((NH4)2SO4) [30]. This suggests that increased NH4+ concentration significantly reduces the expression of OsAMT1.1. Compared with ammonium, nitrate can inhibit the absorption of ammonium salts, resulting in a decrease in the expression of OsAMT genes (OsAMT1;1, OsAMT1;2 and OsAMT1;3) under ammonium–nitrate mixed conditions [31]. Studies in Arabidopsis revealed that ammonium transporters, such as AtAMT1.1 AtAMT1.3 and AtAMT2, are transcriptionally regulated by urea [32]. Therefore, it can be concluded that urea has the ability to affect the activity of AMT genes and the up-regulation of these genes can enhance the plant roots’ ability to uptake NH4+. However, when compared to other crops, there is a lack of research on the AMT gene family in sweet potatoes and the function of IbAMT transporters in NH4+ absorption and utilization. Additionally, it remains unclear whether the expression of IbAMTs is related to sweet potato storage root development and yield under different nitrogen treatments.

To address this gap in knowledge, this study aims to identify the members of the AMT gene family in sweet potatoes and screen for key IbAMT genes that may regulate N utilization and impact storage root development and yield in sweet potatoes. The findings of this study will provide a theoretical basis for further investigation into the function and mechanism of IbAMT regulating N uptake and utilization in sweet potatoes.

(2) I propose shortening the title of the manuscript, it could read, for example: "Genome-wide Analysis of Sweet Potato Ammonium Transporter (AMT): Influence Nitrogen Utilization, Storage Root Development and Yield".

Answer: Thanks! We have revised.

(3) Abstract - both the purpose of the research and the research hypothesis are too poorly outlined here. Please specify the application significance of the research results.

Answer: Thank you for your objective comments. We revised as ‘Abstract: (1) Background: Ammonium as a major inorganic source of nitrogen (N) for sweet potato N utilization and growth, is specifically transported by ammonium transporters (AMTs). However, the activities of AMT family members in sweet potatoes have not been analyzed; (2) Methods: In the present study, the sweet potato cultivar ‘Pushu 32’ which planted in a large area in China was used in field experiments at Agricultural Base of Hainan University (20° 06'N, 110° 33'E) in 2021, and Sanya Nanfan Research Institute of Hainan University (18° 30'N, 109° 60'E) in 2022. Four N levels were tested: 0, 60, 120 and 180 kg ha-1; (3) Results: 12 IbAMT genes were identified in the sweet potato genome, classified into three distinct subgroups based on phylogeny, the same subgroup genes had similar properties and structures. IbAMT1.3 and IbAMT1.5 were mostly expressed in the storage root under N deficiency. Compared with NN and HN, IbAMT1.3 expression and IbAMT1.5 expression, N content in storage root, N uptake efficiency at canopy closure, N fertilization contribution rates, storage root number per plant, storage root weight, and yield of MN were all increased. Furthermore, there was a significant positive correlation between the expression of IbAMT1.3 and IbAMT1.5 with N content in the storage root of sweet potato; (4) Conclusions: IbAMT1.3 and IbAMT1.5 may regulate the N utilization, affect the development of storage root and determine the yield of sweet potato. The results provide valuable insights into the AMT gene family's role in the case of N utilization and affects storage root development and yield in sweet potatoes.’ Lines 19-38.

(4) Keywords - please do not use words that have already appeared in the title of the manuscript. Furthermore, keywords should be arranged in alphabetical order.

Answer: We revised keywords as ‘field experiments; IbAMT1.3; IbAMT1.5; N deficiency; N uptake efficiency’. Lines 39-40.

(5) Introduction - lacks the depth necessary to sufficiently contextualize the research, making it challenging for readers to grasp the significance of the subsequent findings. Lines 75-79 - please outline your research hypothesis better here.

Answer: Thank you for your objective comments. We revised as ‘However, when compared to other crops, there is a lack of research on the AMT gene family in sweet potatoes and the function of IbAMT transporters in NH4+ absorption and utilization. Additionally, it remains unclear whether the expression of IbAMTs is related to sweet potato storage root development and yield under different nitrogen treatments.’ Lines 93-97.

(6) Tables 1, 2, 3 and 4 - explain what the values after the +/- sign mean; instead of "Year * Treatment" give "Year x Treatment".

Answer: We added ‘The values after the ± sign mean standard deviation.’ We have revised.

(7) Figure 1 - explain the symbols NN, LN, MN and HN under the figure.

Answer: We have revised as ‘NN, no N; LN, 60 kg ha-1 N; MN, 120 kg ha-1 N; HN, 180 kg ha-1 N.’

(8) Figure 2 - on the Y axis, instead of "weight" put "Weight of ... " (what?). Explain what the letters on the bars and the symbols on the x-axis mean.

Answer: We have revised as ‘Weight of different organs.’ We added ‘Values followed by different lowercase letters within a group are significantly different among nitrogen treatments. NN, no N; LN, 60 kg ha-1 N; MN, 120 kg ha-1 N; HN, 180 kg ha-1 N.’

(9) Table 3 - in the second heading, replace "Kg" with "kg".

Answer: We have revised.

(10) Lines 114 and 136 - remove "*—p<0.05" - this case is not in the table.

Answer: We have deleted.

(11) Figure 3 - on the Y axis, replace "N" with "N content". Explain the designations NN, LN, MN and HN under the figure. Explain what the letters above the bars mean.

Answer: We added as ‘NN, no N; LN, 60 kg ha-1 N; MN, 120 kg ha-1 N; HN, 180 kg ha-1 N. Values followed by different lowercase letters within a group are significantly different among nitrogen treatments.’

(12) Figures 9 and 10 - explain what the letters above the bars mean.

Answer: We added as ‘Values followed by different lowercase letters within a group are significantly different among nitrogen treatments.’

(13) Line 247 279 - remove "**—p<0.01" - this case is not in the figure.

Answer: Thank you for your objective comments. We have revised.

(14) Line 306 - add "...and Plant Material".

Answer: We changed as ‘Experimental Site and Plant Material’. Line 369.

(15) Line 307 - please add the scientific name of the species.

Answer: We revised as ‘Field experiments were conducted by using sweet potato ‘Pushu 32’ [Ipomoea batatas (L.) Lam.] from October 2021 to February 2022.’ Lines 370-371.

Round 2

Reviewer 2 Report

Comments and Suggestions for Authors

The paper might be accepted in the current form

Comments on the Quality of English Language

The paper might be accepted in the current form